# A new method for calculating average visibility: from the relationship between extinction coefficient and visibility

Zefeng Zhang, Hengnan Guo, Hanqing Kang, Jing Wang, Junlin An, Xingna Yu, Jingjing Lv, Bin Zhu

Key Laboratory for Aerosol-Cloud-Precipitation of China Meteorological Administration, Nanjing University of Information Science & Technology, Nanjing, 210044, China

*Correspondence to*: Zefeng Zhang (zhangzf01@vip.163.com)

**Abstract.** Visibility data are fundamental meteorological observation data widely used in many fields. When using visibility data, it is often necessary to calculate the average visibility, which used to be the

arithmetic average of the visibility data directly. In this study, we first analyze the relationship between the visibility, the extinction coefficient, and atmospheric compositions. Then we propose to use the harmonic average of visibility data as the average visibility, which can better reflect changes in atmospheric extinction coefficients and aerosol concentrations. It is recommended to use the harmonic average visibility in the studies of climate change, atmospheric radiation, air pollution, environmental

health, etc.

## 1 Introduction

Visibility is a fundamental meteorological parameter (WMO, 1957, 2018) and has a wide range of application scenarios. On the on hand, as an indicator of atmospheric transparency, visibility data are used in many aspects of daily life, such as ground transportation (Ashley et al., 2015; Peng et al., 2017),

aviation (Herzegh et al., 2015), and navigation (Debortoli et al., 2019), and in scientific research related to weather processes, such as the study of the formation and dissipation of fog. On the other hand, because visibility ($v$) is determined as a function of the atmospheric extinction coefficient ($b$) at a given contrast threshold ($\varepsilon$) (Koschmieder, 1924) (Eq. 1), and because the extinction coefficient is predominantly determined by aerosol concentrations (Che et al., 2007), visibility can also be used as a parameter

describing atmospheric extinction coefficients (Zhang et al., 2017; Field et al., 2009) and aerosol concentrations (Rosenfeld et al., 2007; Chen et al., 2005), which is widely used in research related to climate change (Rosenfeld et al., 2007; Vautard et al., 2009), atmospheric radiation (Wang et al., 2009;

Wu et al., 2014), atmospheric pollution (Gunthe et al., 2021; Yang et al., 2017) and environmental health (Huang et al., 2009; Laden et al., 2006).

$$v = -\frac{\ln \varepsilon}{b} \qquad (1)$$

A large amount of gridded visibility data have been accumulated through long-term observations at dense measurement sites (Pitchford et al., 2007; Singh et al., 2017), which greatly support many research. Calculating the average visibility is the most frequently performed task when using visibility data (An et al., 2019; Kessner et al., 2013; Zhang et al., 2010). It is easy to see how problems in calculating the average visibility could affect the credibility of the conclusions reached in previous studies using visibility data. Therefore, it is necessary to discuss the method of calculating the average visibility.

There are two variables in Eq. 1, visibility and the extinction coefficient, from which two methods for calculating the average visibility can be derived. The first method directly calculates the arithmetic average of visibility data using Eq. 2, where $\overline{v_2}$ represents the arithmetic average of visibility data, $n$ is the number of measurements, and $v_i$ denotes the visibility obtained in the $i^{th}$ measurement. As can be seen from Eq. 2, the average visibility calculated by the first method is the arithmetic average visibility.

$$\overline{v_2} = \frac{\sum_{i=1}^{n} v_i}{n} \qquad (2)$$

The second method calculates the average extinction coefficient data first, then substitutes the average extinction coefficient into Eq. 1 to obtain the average visibility; the specific derivation process and results are shown in Eq. 3. Specifically, first, substitute the visibility measurement $v_i$ into Eq. 1 to obtain the corresponding extinction coefficient $b_i$ in the $i^{th}$ measurement. Then, calculate the arithmetic average of a total of $n$ extinction coefficients, denoted as $\overline{b}$. Finally, substitute the average extinction coefficient into Eq. 1 to obtain the average visibility $\overline{v_3}$. As can be seen from Eq. 3, the average visibility calculated by the second method is the harmonic average visibility.

$$b_i = \frac{\ln \varepsilon}{v_i} \implies \overline{b} = \frac{\sum_{i=1}^{n} b_i}{n} = -\frac{\sum_{i=1}^{n} \frac{\ln \varepsilon}{v_i}}{n} = -\frac{\ln \varepsilon}{n} \sum_{i=1}^{n} \frac{1}{v_i} \implies \overline{v_3} = -\frac{\ln \varepsilon}{\overline{b}} = \frac{n}{\sum_{i=1}^{n} \frac{1}{v_i}} \qquad (3)$$

Equation 2 gives the arithmetic average visibility and Eq. 3 gives the harmonic average visibility. It is clear that the values of average visibility calculated by the two methods are different. This is because atmospheric visibility is constantly changing, and it has been mathematically proven that, unless all values used to calculate the average are the same, the arithmetic average is always greater than the harmonic average (Ferger, 1931).

The question arises as to whether the average visibility used in practical work should be the arithmetic average visibility calculated by Eq. 2 or the harmonic average visibility calculated by Eq. 3. To date, arithmetic average visibility has been used in studies (An et al., 2019; Kessner et al., 2013; Rosenfeld et al., 2007; Singh et al., 2017; Zhang et al., 2017) and harmonic average visibility has never been an option, so that when studies refer to average visibility, it is calculated directly using Eq. 2 without the need for clarification. The answer seems clear, but not yet convincing. This is because no theoretical justification has been given in past studies for using the arithmetic average visibility rather than the harmonic average visibility. Although it is true that the arithmetic average visibility is more intuitive, this does not exclude the possibility that the option of the harmonic average visibility has been overlooked in the past due to the blind spot in thinking. Therefore, a more in-depth discussion is necessary.

The first thing to do is to compare the difference in numerical values of the average visibility obtained by the two methods. If the difference is negligible, there is no point in discussing this issue, and the arithmetic average visibility obtained from Eq. 2 is also reliable. However, if the difference is considerable, it is necessary to analyze the difference in physical meaning between arithmetic average visibility and harmonic average visibility, and then select the appropriate calculation method for average visibility in different scenarios in combination with the purpose of using visibility data.

**2 The numerical difference between arithmetic average visibility and harmonic average visibility**

To develop an intuitive understanding of the magnitude of the numerical difference between arithmetic average visibility and harmonic average visibility, we analyzed the visibility data measured at 1-min resolution by a CJY-1 visibility meter (CAMA Measurement & Control Equipments Co., Ltd) on the campus of the Nanjing University of Information Science and Technology in Nanjing, China,

during 2010–2017. The details regarding the observation site and instruments are given in Zhang et al.

(2017).

80       The hourly, daily, monthly, and yearly arithmetic average visibility and harmonic average

visibility are shown in Fig. 1a and 1b, respectively. By substituting the values of average visibility

during the corresponding period shown in Fig. 1a and Fig. 1b into Eq. 4, we obtain the relative

deviation of the hourly, daily, monthly, and yearly arithmetic average visibility from harmonic average

visibility. Figure 1c shows the distribution of the magnitude of relative deviation. The value of 96.3 in

the lower-left corner of Fig. 1c indicates that 96.3% of the relative deviation of the hourly average

visibility falls within the range of 0–10%.

$$X\% = \frac{\overline{v_2} - \overline{v_3}}{\overline{v_3}} \times 100\%$$

(4)

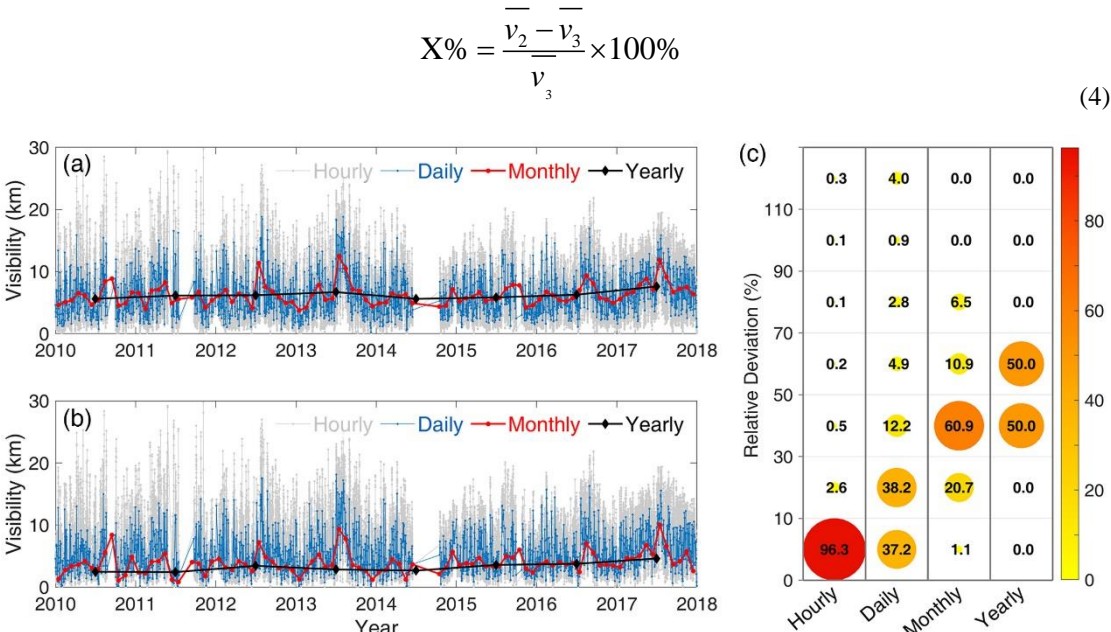

**Figure 1: Comparison of arithmetic average visibility and harmonic average visibility: (a)**

**arithmetic average visibility calculated using Eq. 2. (b) harmonic average visibility calculated using**

**Eq. 3. (c) distribution of the relative deviation of arithmetic average visibility from harmonic**

**average visibility.**

      As shown in Fig. 1, the arithmetic average visibility calculated using Eq. 2 (Fig. 1a) is always

higher than the harmonic average visibility calculated using Eq. 3 (Fig. 1b); therefore, all values of the

relative deviation lie in the range of greater than zero. The results in Fig. 1 are not a coincidence

because of the specificity of the measurement data, but an inevitable result that will appear when

calculating the average of any visibility measurement data using Eq. 2 and Eq. 3. It has been

mathematically proven that, unless all values used to calculate the average are the same, the arithmetic average is always greater than the harmonic average; the greater the variation in the data, the greater

the difference between the two.

The relationship between the arithmetic average and the harmonic average can explain the distribution of relative deviation values in Fig. 1c. The range of the measured visibility values is typically related to the observation period. The longer the duration of the observation, the larger the range of the measured visibility data. Therefore, the longer the observation period chose to calculate

the average visibility, the larger the relative deviation of the arithmetic average visibility from the harmonic average visibility. It is not difficult to understand why the relative deviation of the yearly average is larger than that of the monthly average, which is larger than that of the hourly average, according to the distribution of the relative deviation shown in Fig. 1c.

Regarding the relative deviation of yearly and monthly arithmetic average visibility from

harmonic average visibility (Fig. 1c), most of the values fall within the range of 30% to 70%, which is far greater than the typical range of measurement error of visibility meters (WMO, 2018). Regarding the relative deviation of hourly and daily average visibility, although most of the values are less than 30%, this does not mean that the difference between the arithmetic average and the harmonic average can be ignored. Because atmospheric visibility can sometimes change significantly in a short time, a

topic of particular interest in previous studies, at which time the average visibility calculated by the two methods can be quite different.

In summary, as long as the atmospheric visibility is variable, the values of arithmetic average visibility and harmonic average visibility will not be the same, and the magnitude of the difference between them is related to the intensity of the change in visibility. Therefore, the difference between

the two calculation methods cannot be ignored in large-scale and long-term studies. Even for small-scale and short-term studies, the difference is not negligible when there is a significant change in visibility.

**3 Discussion of the physical meaning of the two calculation methods of average visibility**

**3.1 Discussion of the extinction coefficient and visibility**

To understand the difference in physical meaning between arithmetic average visibility and harmonic average visibility, it is necessary to understand the characteristics of the two physical quantities, extinction coefficient and visibility. To this end, we design a thought experiment.

Assume there is a system, where there are a total of $n$ substances affecting the extinction coefficient. The mass concentration of the $i^{\text{th}}$ substance is $m_i$ and the mass extinction coefficient is $M_i$.

We carry out a thought experiment, and the experimental procedures and corresponding results are recorded in Table 1.

**Table 1. Records of the thought experiment process**

| Experimental procedure | Extinction coefficient | Visibility |
|---|---|---|
| 1. Remove all substances from the system | 0 | $+\infty$ |
| 2. Add the first substance to the system | $m_1 M_1$ | $-\dfrac{\ln \varepsilon}{m_1 M_1}$ |
| 3. Continue adding the second substance to the system | $m_1 M_1 + m_2 M_2$ | $-\dfrac{\ln \varepsilon}{m_1 M_1 + m_2 M_2}$ |
| 4. Continue adding the $i^{\text{th}}$ substance to the system | $m_1 M_1 + m_2 M_2 + \cdots + m_i M_i$ | $-\dfrac{\ln \varepsilon}{m_1 M_1 + m_2 M_2 + \cdots + m_i M_i}$ |
| 5. Repeat the above until all $n$ substances are added to the system | $\displaystyle\sum_{i=1}^{n} m_i M_i$ | $-\dfrac{\ln \varepsilon}{\displaystyle\sum_{i=1}^{n} m_i M_i}$ |

Two conclusions can be drawn from the results of the thought experiment recorded in Table 1. It should be noted that these two conclusions are not new knowledge but the basis for subsequent

discussion.

The first conclusion is that the concentration and the optical properties of the substances determine the extinction coefficient and the visibility of the system. This suggests that the changes in

the extinction coefficient and visibility of the system should logically match the changes in the mass concentration and mass extinction coefficient of the substances in the system.

The second conclusion is that the extinction coefficient is an extensive quantity, whereas the visibility is neither an extensive nor an intensity quantity. This is because the extinction coefficient is proportional to the amount of matter in the system, suggesting that the extinction coefficient is an extensible quantity. The visibility decreases as the amount of matter in the system increases, suggesting that visibility is not an extensible quantity. The magnitude of visibility varies with the concentration of

the substance in the system, so it is not a characteristic property of the substance and not an intensity quantity. Therefore, the summation of visibility has no real physical meaning.

**3.2 Discussion on the physical meaning of arithmetic average visibility and harmonic average visibility**

Simulated measurements are generated in order to discuss the physical meaning of arithmetic

average visibility and harmonic average visibility. Assuming that a total of $n$ measurements are made with the same instrument, at the same site, at the same time interval, and the measurement results are considered reliable, Eq.5 relates the mass concentration ($m_j$) and the mass extinction coefficient ($M_j$) of substances to the extinction coefficient, and to the visibility in the $j^{th}$ observation.

$$M_j m_j = b_j = -\frac{\ln \varepsilon}{v_j} \tag{5}$$

Then we calculate the average extinction coefficient and average visibility with three methods, respectively.

Method 1. Based on the first conclusion in section 3.1, the average extinction coefficient and average visibility were calculated using the concentrations and optical properties of the substances during the observation period, as the definition implies. First, calculate the average mass concentration

and the average mass extinction coefficient during the observation period, as shown in Eq. 6. Then, calculate the average extinction coefficient and average visibility using the average mass concentration and the average mass extinction coefficient during the observation period, as shown in Eq. 7.

$$\overline{m} = \frac{\sum\limits_{j=1}^{n} m_j}{n}, \quad \overline{M} = \frac{\sum\limits_{j=1}^{n} M_j m_j}{\sum\limits_{j=1}^{n} m_j} \tag{6}$$

$$\overline{b} = \overline{m}\,\overline{M} = \frac{\sum\limits_{j=1}^{n} M_j m_j}{n}, \quad \overline{v} = -\frac{\ln \varepsilon}{\overline{b}} = -\frac{\ln \varepsilon}{\overline{m}\,\overline{M}} = -\frac{n \ln \varepsilon}{\sum\limits_{j=1}^{n} M_j m_j} \tag{7}$$

Method 2. Substitute the observed mass concentration and mass extinction coefficient into Eq. 2 to obtain the arithmetic average visibility, which is then substituted into Eq. 1 to obtain the corresponding average extinction coefficient, as shown in Eq. 8.

$$\overline{v_2} = \frac{\sum\limits_{j=1}^{n} v_j}{n} = \frac{\sum\limits_{j=1}^{n} \dfrac{-\ln \varepsilon}{b_j}}{n} = -\frac{\ln \varepsilon}{n} \sum\limits_{j=1}^{n} \frac{1}{M_j m_j}, \quad \overline{b_2} = -\frac{\ln \varepsilon}{\overline{v_2}} = \frac{n}{\sum\limits_{j=1}^{n} \dfrac{1}{M_j m_j}} \tag{8}$$

Method 3. Substitute the observed mass concentration and mass extinction coefficient into Eq. 3 to obtain the harmonic average visibility, which is then substituted into Eq. 1 to obtain the corresponding average extinction coefficient, as shown in Eq. 9.

$$\overline{v_3} = \frac{n}{\sum\limits_{j=1}^{n} \dfrac{1}{v_j}} = \frac{n}{\sum\limits_{j=1}^{n} \dfrac{b_j}{-\ln \varepsilon}} = -\frac{n \ln \varepsilon}{\sum\limits_{j=1}^{n} b_j} = -\frac{n \ln \varepsilon}{\sum\limits_{j=1}^{n} M_j m_j}, \quad \overline{b_3} = -\frac{\ln \varepsilon}{\overline{v_3}} = \frac{\sum\limits_{j=1}^{n} M_j m_j}{n} \tag{9}$$

The average visibility and average extinction coefficient calculated by the three methods are now compared and analyzed. A comparison of Eq. 7 and Eq. 9 indicates that the expression of the average visibility and the expression of the average extinction coefficient are identical respectively, while the expressions in Eq.8 are different from those in Eq. 7 and Eq. 9.

The reason for this can be explained by the second conclusion given in Section 3.1. All three methods perform summation. The Method 1 and Method 3 both carry out the summation over extensive quantities, i.e. the mass and the extinction coefficient, so that their corresponding physical meanings are clear. The Method 1 and Method 3 actually describe the same physical process, i.e. the mixing process. However, the Method 2 carries out the summation over the visibility, which is neither an extensive quantity nor an intensity quantity, so that the results of the summation of visibility are just

numerical values with no corresponding physical process. Therefore, the arithmetic average visibility has no real physical meaning.

The difference in the physical meaning of arithmetic average visibility and harmonic average visibility leads to the difference in the derived expressions of the average extinction coefficient. It can be seen from Eq.7 and Eq.9 that the expression of the average extinction coefficient derived from the harmonic average visibility (Eq.9) is identical to that derived from the definition of the extinction coefficient (Eq.7). However, a comparison of Eq. 7 and Eq. 8 indicates that the expression of the

average extinction coefficient derived from the arithmetic average visibility (Eq.8) differs from that derived from the definition of the extinction coefficient (Eq.7). This suggests that we should use the harmonic average visibility rather than the arithmetic average visibility when using average visibility data to obtain average extinction coefficient. Considering that the main contribution to atmospheric extinction comes from aerosol particles, it is also appropriate to use harmonic average visibility data for

research on aerosols using visibility data.

In summary, if the purpose is to numerically describe the measured visibility data, then the calculation of average visibility can be treated as a mathematical problem, and the arithmetic average visibility can be used to represent the average visibility. However, if the average visibility is used as a parameter to characterize changes in atmospheric extinction coefficients and aerosol concentrations,

especially in research related to climate change, atmospheric radiation, atmospheric pollution, environmental health, etc., then the calculation of average visibility should be treated as a physical problem, and the harmonic average visibility should be used to represent the average visibility.

**4 Conclusions**

This study proposes a new method for calculating the average of visibility data, i.e. harmonic

average visibility. The main differences between the proposed harmonic average visibility from the previously used arithmetic average visibility are as follows.

1. The numerical values of harmonic average visibility and arithmetic average visibility are different. The values of harmonic average visibility are always smaller than the corresponding arithmetic average visibility, and the difference between them becomes larger as the observed visibility

values fluctuate more strongly. Therefore, the method for calculating the average visibility should be

carefully selected when analyzing large-scale or long-term visibility data, and when analyzing local

visibility data with large changes in visibility within a short period of time.

2. Compared to the arithmetic average visibility, the harmonic average visibility can better represent

changes in average atmospheric extinction coefficients and average aerosol concentrations. Therefore,

we recommend preferentially using harmonic average visibility when calculating the average of visibility

data in research related to climate change, atmospheric radiation, atmospheric pollution, environmental

health, etc.

**Acknowledgements**

This work was supported by the National Key Research and Development Program of China (No.

2019YFC0214604). We thank WeiWei Wang, Li Xia and Jiade Yan for offering visibility data and

maintaining the observation site used in this study.

**Data availability**

The       data       supporting       the       conclusions       have       been       deposited       in       Zenodo

(https://doi.org/10.5281/zenodo.5025882).

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
