# Peer review of "A new method for calculating average visibility: from the relationship between extinction coefficient and visibility"

_Atmospheric Measurement Techniques, 2022_

## Author Comment (AC2)

**Response to reviewers for manuscript AMT-2022-74: Average visibility that has been miscalculated**

We appreciate the editorial team and the reviewers for their time and comments towards improving our manuscript. Considering the relevance of two reviewers' comments, we respond to all of the points together below.

**Anonymous Referee #1**

The authors proved that the algorithm previously used to estimate average visibility is incorrect, and it causes the problems in previous studies related to reliability and credibility. I don't think it is a 'correct' method to calculate the average of the visibility and other methods in the previous studies were incorrect. In this study, I wonder what the definition of the terms 'correct' and 'incorrect' is. I prefer to utilize the term 'an improved method.' In my opinion, it is one of the scientific methods for investigating the natural world instead of the absolutely correct way to estimate the average visibility.

**Anonymous Referee #2**

This study claims to provide correct algorithm for calculating average visibility. The authors also argued that the methods used in previous studies were not correct.

First of all, I don't agree with the title of the paper. Saying "Average visibility that has been miscalculated" is completely wrong. There is no evidence that previous studies miscalculated visibility, so the paper's title is misleading. I agree with reviewer 1, making the claim that this study provides a correct method is not appropriate, while it may be one of the methods to calculate visibility.

The message of the current manuscript is misleading, therefore it should not be considered for publication. The authors should rewrite the manuscript by proposing their algorithm as one of the methods of calculating visibility. However, again it is a question of how reliable the proposed method is. If the authors take the issue raised carefully and resubmit the manuscript, I can review it again for any possibility of publication in the AMT.

**General response:**

For ease of discussion, we have grouped the two reviewers' comments into three questions to reply, in which the responses to questions 1 and 2 are more focused on the comments of Reviewer 1, and the responses to questions 2 and 3 are more focused on the comments of Reviewer 2.

Question 1: Is this a change from "incorrect" to "correct" or a "general improvement"?
 Response 1: This is a very important question, because the answer to this question is directly related to the evaluation of the value of this manuscript.

First of all, we agree that there may be no absolute correctness in the world in an absolute sense, and therefore all research work can only be "improvements" rather than changes from "incorrect" to "correct". However, people do not discuss issues in an absolute sense in specific work, otherwise the word "correct" would have no value. We believe that some improvements can be called changes from "incorrect" to "correct"

according to the content of improvements, while some are just general improvements. In order to clarify the difference between the two, we will start with an example for a detailed explanation.

Example: A car is travelling on a road. The average speed of the car is measured to be  $v_1$ ,  $v_2$  and  $v_3$  when travelling uphill, on a flat road and downhill respectively. What is the average speed of the car  $(\bar{\nu})$ ?

Student A first proposed the first method to calculate the average speed, as shown in Eq. 1.

$$\bar{v} = \frac{v_1 + v_2 + v_3}{3} \tag{1}$$

Student B thought that the measurement error of the speed of the car is related to the slope and should be corrected. Therefore, student B suggested that the average speed should be calculated using Eq. 2, where  $c_1$ ,  $c_2$  and  $c_3$  are the correction factors.

$$\overline{v} = \frac{c_1 v_1 + c_2 v_2 + c_3 v_3}{3} \tag{2}$$

Student C thought that student A had misunderstood the concept of speed, and that the correct average speed should be calculated by dividing the total distance travelled by the time taken, as shown in Eq. 3, where  $t_1$ ,  $t_2$  and  $t_3$  correspond to the times the car runs at speeds of  $v_1$ ,  $v_2$  and  $v_3$  respectively, and t is the total running time of the car.

$$\overline{v} = \frac{t_1 v_1 + t_2 v_2 + t_3 v_3}{t} = \frac{t_1}{t} v_1 + \frac{t_2}{t} v_2 + \frac{t_3}{t} v_3$$
(3)

We think that Eq. 2 and Eq. 3 are both improvements to Eq. 1, but Eq. 2 is only a general improvement, whereas Eq. 3 is an improvement from "incorrect" to "correct". This is because the improvement of Eq. 3 corrects the misunderstanding of the concept of the average speed in Eq. 1 and clearly points out the cause of the error, that is, the "weight" of the values should be considered when calculating the average value. However, the improvement of Eq. 2 does not improve the perception of the concept and is a technical improvement.

The improvement of the proposed algorithm for average visibility to the old algorithm is identical in nature to the improvement of Eq. 3 to Eq. 1. The proposed algorithm is derived considering the "weight" of the values when calculating the average visibility, whereas the old one does not. This improvement is not a technical one, but rather a cognitive one, and we therefore consider our improvement a change from "incorrect" to "correct".

**2. Question 2:** Why do you think that the new algorithm is "correct" and the old one is "incorrect"?

**2. Respond 2:** We have presented a rigorous proof in the manuscript. Here we use an extreme example to illustrate why the new algorithm is "correct" and the old one is "incorrect".

Suppose there are two kinds of boxes of the same volume, box A is filled with gases and aerosols with a horizontal visibility of v, and box B is a perfect vacuum so that the

visibility is infinite. We mix uniformly a certain number of boxes A with boxes B, and then calculate the average visibility after mixing using the new algorithm and the old one, respectively, the results of which are given in Table R1 and Table R2.

First, we mix one box B with a different number of boxes A. The average visibility calculated using the new algorithm and the old algorithm is given in Table R1. It can be seen from Table R1 that as the number of boxes A increases, the average visibility after mixing calculated by the new algorithm gradually converges to the visibility of box A, while the average visibility calculated by the old algorithm is always infinite. Then, we mix one box A with a different number of boxes B. The average visibility calculated by two algorithms is given in Table R2. It can be seen from Table R2 that as the number of boxes B increases, the average visibility calculated by the new algorithm gradually converges to the visibility of box B, while the average visibility calculated by the old algorithm remains infinite. Clearly, the results calculated by the new algorithm are more reasonable than the results of the old algorithm. The difference between the old and new algorithms is essentially a matter of the weight of the values of observed visibility data. The visibility is determined by the extinction coefficient of the medium through which the light propagates, so the weight should match the extinction coefficient of the medium when calculating the average of visibility data. Large weighting factors should be given to the relatively small visibility values corresponding to the large extinction coefficient. But the old algorithm is the opposite, giving large weighting factors to those large visibility data corresponding to relatively small extinction coefficients.

| Number of box A | Average visibility calculated | Average visibility calculated |
|-----------------|-------------------------------|-------------------------------|
|                 | by the new algorithm          | by the old algorithm          |
| 1               | v /2                          | $\infty +$                    |
| 2               | 2 v/3                         | $\infty +$                    |
| 3               | 3 v/4                         | $\infty +$                    |
| 4               | 4 v/5                         | $\infty +$                    |
|                 |                               |                               |
| n               | nv / (n+1)                    | $\infty +$                    |

Table R1. The average visibility calculated by the new algorithm and the old one when one box B is mixed with a different number of boxes A.

| Table R2. The average visibility calculated by the new algorithm and the old one | when |
|----------------------------------------------------------------------------------|------|
| one box A is mixed with a different number of boxes B.                           |      |

| Number of box B | Average visibility calculated | Average visibility calculated |
|-----------------|-------------------------------|-------------------------------|
|                 | by the new algorithm          | by the old algorithm          |
| 1               | 2v                            | $\infty +$                    |
| 2               | 3v                            | $\infty +$                    |
| 3               | 4v                            | $\infty +$                    |
| 4               | 5v                            | $\infty +$                    |
|                 |                               |                               |
| n               | nv                            | $\infty +$                    |

**3.** Question 3: Discussion of the relationship between the evidence and the conclusion. **3.** Response 3: The argument that "there is no evidence that previous studies miscalculated visibility" does not lead to the conclusion that the algorithm for calculating the average visibility in the past is correct, nor to the conclusion that the title of the manuscript is misleading. This is because in many cases people only look for evidence when they realize that there exists a problem. A well-known example is that before Galileo, it was a common belief that "heavier objects fell faster than lighter ones". No one could give conclusive evidence denying the above conclusion at that time until Galileo's thought experiment.

Returning to the issue of the algorithm for average visibility in this manuscript, we think that we should not decide that the old algorithm is correct and then come to reject the new algorithm from the start, but rather should look at the process of proving the algorithm to determine which is correct. However, the commonly used old algorithm has not been rigorously verified, which probably has been neglected in past research. Instead, we not only present the new algorithm for average visibility, but also prove that the new algorithm is correct and the old one is incorrect. The rigorous proof is presented in the manuscript. In brief, the weight should be considered when calculating the average. The visibility is determined by the extinction coefficient of the medium through which the light propagates. Therefore, the weight should match the extinction coefficient of the medium when calculating the average of visibility data. The answers to Question 1 and Question 2 in this response can help to understand the difference between the old and new algorithms, i.e., the new algorithm considers the weight of the values of observed visibility data, whereas the old one does not. If we cannot find a problem in the process of proving, we should conclude that the new algorithm and the old algorithm cannot be correct at the same time, and the new algorithm is the correct one.

To summarize, neither of the two reviewers denied the significance of discussing the algorithm for average visibility, and did not raise any objections to the proof process of the new algorithm in the referee comments. In other words, the two reviewers did not object to the manuscript at a substantive level, but actually expressed doubts about the conclusions of the manuscript out of caution or habitual thinking. We hope that this response will dispel the doubts of the two reviewers.

[revised manuscript text omitted]

---

## Referee Report (RR1)

**Review of "Average visibility that has been miscalculated"**

Zhang et al. report a reassessment of the equations used to calculate the average visibility and propose that the usual understanding of average visibility be dropped in favour of a weighted average. Their work highlights the important connection between the visibility and the underlying atmospheric extinction coefficient, which itself is connected to the composition of the atmosphere (particle concentration, composition, hygroscopicity, etc). Users of visibility data need to be alert to the differences between Eqs. (2) and (3), and that average visibility cannot be straightforwardly related to average extinction coefficient.

The authors do well to bring this to the attention of the community, but their approach has several flaws that should be addressed. I do not think that the work is publishable in its present form. It is not simply a question of the title, an issue raised by the two reviewers. The authors need to revise and clarify the argument and points made in the manuscript.

A major objection is that "average visibility" (Eq. 2) conforms to the accepted definition of arithmetic average or arithmetic mean of a given property, and this is what most scientists would understand by the term. The authors are right that Eq. (2) and (3) are not mathematically equivalent. What is needed then is for the authors to provide helpful terminology distinguishing between Eq. (2) ("average visibility") and Eq. (3). There are presumably circumstances when researchers would find the average visibility (Eq. 2) a useful concept for describing their observations. In other cases, a weighted average or some other statistic would be more appropriate (using Eq. (3) to study the underlying extinction coefficient). The circumstances for using one or other statistic needs to be clarified.

Because it may be reasonable to use one or other equation depending on a study's goals, the manuscript's terminology denoting logical conclusions ("therefore", "proves"), and correctness ("correct", "miscalculated", "error") is too narrow. This paper would be much more valuable to the scientific community if it brought greater clarity and nuance to the ways in which visibility data are analysed.

More emphasis should be placed on when the values produced by the two equations differ. The values produced by Eq. (2) and (3) converge as the range of visibility values becomes increasingly narrow. As Fig. 1c shows, the variation in the results of the two approaches within the hourly dataset is small, presumably because visibility is generally changing little over this time period for most observations. Greater variation occurs when datasets are long enough to contain larger variations in the individual visibility measurements.

The manuscript's arguments about the average value becoming infinite if any one of the measurement series is infinite are not convincing. Such contributions to the average visibility (or another physical property) do not occur in practice because individual measurements giving infinity are physically implausible (whether for visibility or another physical property) and would be removed from the data set. The formal mathematical possibility of an infinite result is not helpful.

The manuscript title is uninformative and potentially misleading. Both reviewers thoroughly disliked the title, but the authors were resistant to changing it. I think they should. Perhaps something like "Average visibility and its relationship to atmospheric extinction: a clarification" provides a better summary of their paper's content and aims.

---

## Referee Report (RR2)

**A new method for calculating average visibility: from the relationship between extinction coefficient and visibility**

Zhang et al. address the question of the calculation and interpretation of the average visibility and its relationship to the extinction coefficient. While the equation relating visibility and extinction is straightforward, the use of a simple arithmetic average of visibility can be misleading, producing a statistic that is disconnected from the underlying physical cause of visibility reduction, namely, atmospheric extinction. Researchers therefore need to be aware of these considerations if they use visibility datasets to investigate underlying trends in the atmospheric extinction.

This is a revised manuscript that addresses many of the deficiencies of the earlier submission. There some weaknesses in the manuscript that should be addressed to improve its clarity. Some of these are minor textual changes while others require greater clarification. These are addressed below.

L1.: Omit colon in title

L18: "On the ONE hand"

L32: Rephrase "which greatly support many research." Unclear.

L34: Preferable: "Methodological issues in calculating the average visibility..." or similar L43: Given that the paper focuses on different average statistics, these should be specified explicitly...."The second method first calculates the arithmetic average of the extinction coefficient,". And likewise elsewhere in the manuscript.

L61: The sentence "The answer seems clear, but not yet convincing." adds little to the argument and should be dropped.

L65: Better: "been overlooked in the past."

L68: "If the difference...is also reliable". The authors have already pointed out that arithmetically the two methods are not the same, while "reliability" relates to the purpose that the data is being used for and the accuracy and precision needed. Some clarification is needed here.

L74: It seems what the authors are doing is not developing an "intuitive understanding", but rather demonstration the divergence of the two approaches with a given dataset.

Fig. 1: It is difficult to compare the short term effects and the two different methods for calculating the average visibility since they are on separate graphs. It would be helpful to have plots (e.g., monthly, yearly) on the same figure to improve comparison.

L104: Correct "period CHOSEN to"

L143: I presume "extensive" property is what is meant?

L146: "the summation of visibility has no real physical meaning". That might be so, yet it could still be a useful statistic.

L147: "3.2 Physical meaning of arithmetic average visibility and harmonic average visibility" is a clearer and shorter section title.

L154/5: This section is confusing as M\_j here relates to the average mass extinction of the SAMPLE, not individual SUBSTANCES composing the sample, which is the meaning for the same symbol in Table 1. This should be clarified in the text and the properties and symbols distinguished from one another.

L158: Change "substances" to "sample" as in the above.

L183: Perhaps "property" is better than "process".

L197: I think this is a statistical rather than a "mathematical" problem.

---

## Author Response (AR2)

Response to reviewers for manuscript AMT-2022-74: **Average visibility that has been miscalculated**

We appreciate the editorial team and the reviewers for their time and comments towards improving our manuscript. We have revised the manuscript carefully, and we respond to all of the reviewers' points below. Responses are given in red.

**Anonymous Referee #1**

**General Comments:**

The authors divided the concerns of the reviewers into three questions and answered them by means of some examples. However, the authors fail to improve the manuscript as suggested by the reviewers. The claim that the previous methods were wrong, is not satisfactory! Detailed comments on the authors' responses are below.

**General Response:** Many thanks to the reviewer for the helpful comments. This time we have made substantial changes to the manuscript to address the reviewers' comments. Specifically, we restrict our focus to proposing a new method for calculating average visibility, rather than dwelling on proving that the old method is incorrect and the new method is correct. We have revised the title, the structure, the argumentative process, and the conclusion of the manuscript. Due to the number of changes, we did not highlight the revised sentences with a different colour. Detailed responses to the comments are below.

**1. Questions 1:** The authors tried to explain through a car travel example that "is this a change from "incorrect" to "correct" or a "general improvement". I don't believe this is the case for visibility calculations and the example given nowhere proves that the previous visibility calculations were incorrect. How did the authors come to the conclusion that the all previous visibility calculations were similar to those in Equation 1 and their Equation 3?

**Response:** In accordance with the reviewer's comments, we no longer use the words "incorrect" and "correct" to describe the differences between the old and new calculation methods in the revised manuscript. Here, as a discussion, we explain our understanding of the difference between "a change from incorrect to correct" and "a general improvement", and how our thinking changed.

We think that if the previously used method contradicts basic scientific principles, and the new method corrects this contradiction, then it is "incorrect" to "correct". If the old method does not contradict basic scientific principles, and the new method leads to a better result, then it is a "general improvement".

The key question here is whether calculating the average of visibility data is a mathematical or physical problem. We have always considered it a physical problem. And we think a physical problem should have its physical meaning and corresponding process. We tried to use the car travel example in our last response to show that the previously used calculation method for average visibility was flawed in a physical sense, but we had not made that clear.

The previously used method carries out the summation over the visibility when

calculating the average of visibility data. The visibility is neither an extensive quantity nor an intensity quantity, so that the results of the summation of visibility are just numerical values with no corresponding physical process. Therefore, the arithmetic average visibility calculated by the previously used method has no real physical meaning, leading to problems in the average extinction coefficient obtained using arithmetic average visibility. The revised manuscript elaborates on this point. On this basis, the new calculation method is justified from the point of view of physical processes, and the argument continues to explain why the harmonic average visibility calculated by the new method better reflects the changes in average atmospheric extinction coefficients and aerosol concentrations.

If the calculation of average visibility can only be understood as a physical process, then the previously used method is considered to be "incorrect" because of the progress without real physical meaning, and the proposed method is considered to be "incorrect" to "correct" compared to the old one. However, in retrospect, if the purpose is to numerically describe the measured visibility data, then the calculation of average visibility can be treated as a mathematical problem; therefore, it cannot be simply assumed that the previously used method is "incorrect".

Therefore, the expressions "correct" or "incorrect" are not used in the revised manuscript. We turn to emphasize that it is important to select the appropriate method for calculating the average visibility when the average visibility is used as a parameter to characterize changes in atmospheric extinction coefficients and aerosol concentrations.

**2. Questions 2:** I agree mathematically their example may be correct but does this example always apply in a real situation where atmospheric visibility cannot be infinite? Therefore, this example does not prove that the previous methods were wrong. I believe previous studies considered atmospheric conditions to calculate visibility and there is always room for improvement in the method depending on the problem and situation.

**Response:** We agree that an example can illustrate the existence of a problem, but it cannot be the basis for a conclusion. We revised the manuscript and give a theoretical derivation to obtain the conclusion. In brief, Eq. 1 gives the expression of the average extinction coefficient derived using previously used arithmetic average visibility, and the detailed derivation of which is given in the manuscript. The extinction coefficient is the product of the mass concentration and the mass extinction coefficient of the substance, as the definition implies. However, the expression of the extinction coefficient in Eq. 1 contradicts the definition of the extinction coefficient. Equation 2 gives the expression of the average extinction coefficient derived using the proposed harmonic average visibility, and the detailed derivation of which is given in the manuscript. The expression of the extinction coefficient in Eq. 2 is exactly in line with the definition of extinction coefficient, showing that the conclusion is reasonable.

$$\overline{b_2} = -\frac{\ln \varepsilon}{\overline{v_2}} = \frac{n}{\sum\limits_{j=1}^{n} \dfrac{1}{M_j m_j}} \tag{1}$$

$$\overline{b_3} = -\frac{\ln \varepsilon}{\overline{v_3}} = \frac{\sum_{j=1}^{n} M_j m_j}{n} \tag{2}$$

We agree with the reviewer that an improvement on Eq. 1 would give better results. But in this case, the direct use of the new method for average visibility is probably a better choice.

**3. Questions 3:** I do not agree with the author's statement - "The argument that "there is no evidence that previous studies miscalculated visibility" does not lead to the conclusion that the algorithm for calculating the average visibility in the past is correct, nor to the conclusion that the title of the manuscript is misleading." Without any scientific evidence how can the author claim that all the previous methods were wrong? I liked the philosophical statement but it doesn't add anything to support their conclusion.

**Response:** The point we were trying to make at the time was that there are no conclusions disproving our point of views at the philosophical level, so the discussion should focus on the specific physical process and corresponding argumentative process.

**4. Comment 1:** Introduction section lacks background and references that should be addressed.

**Response:** Suggested changes have been implemented in the manuscript, where the concluding statement has been removed, underlying ideas for proposing the method is provided, and a description of the differences between the two methods has been added.

**5. Comment 2:** Inferences – I don't understand how the differences in equation 7 from equations 6 prove that the algorithm of equation 2 is wrong. It's just a case of two different ways of calculation. Here authors need solid justification.

**Response:** Thanks for pointing out this. It is true that the differences in equation 7 from equation 6 do not prove that the algorithm of equation 2 is wrong. There is a mistake in the sentence, and it should have been the comparison of Eq. 8 and Eq. 6 in the original manuscript, i.e., the Eq. 1 and Eq. 2 in the response to the Question 1 in this response to the reviewer. The expression in Eq. 1 contradicts the definition of the extinction coefficient, while the expression in Eq. 2 is in line with the definition of the extinction coefficient; therefore, Eq.2 is the reasonable calculation method. The corresponding text has been revised, please refer to the revised manuscript (P. 9, Line 185-195).

**6. Comment 3:** I agree that visibility mainly depends on particles, although studies have shown that gases have non-negligible (about 4-10%) contribution to visibility that needs to be considered.

**Response:** Thanks for pointing out this. Considering that the gas component also has a mass and a mass extinction coefficient, there is no need to discuss it separately and the gas component has been taken into account in the revised manuscript.

**7. Comment 4:** Relative error caused by the erroneous algorithm – This section needs more justifications. Simply, visibility can be calculated using the extinction coefficient at the individual point or the other way around. Furthermore, rather than being called an error, it shows more of a difference in values from methods 2 and 3. Here authors should provide clear and detailed descriptions of equation 9 and results. More experimental (mathematical) examples can help in this direction.

**Response:** Thanks for pointing out this. We now use the word "relative deviation" rather than "relative error" to describe the difference between the two methods. The section describing the relative deviation has been moved to an earlier position to illustrate the differences between the two methods.

**8. Comment 5:** Conclusions – This section needs some improvement. Authors are repeating the same message over and over. They should include possible limitations and improvements for both methods.

**Response:** Suggested changes have been implemented in the manuscript We have revised the conclusions, which now reads "This study proposes a new method for calculating the average of visibility data, i.e. harmonic average visibility. The main differences between the proposed harmonic average visibility from the previously used arithmetic average visibility are as follows.

1. The numerical values of harmonic average visibility and arithmetic average visibility are different. The values of harmonic average visibility are always smaller than the corresponding arithmetic average visibility, and the difference between them becomes larger as the observed visibility values fluctuate more strongly. Therefore, the method for calculating the average visibility should be carefully selected when analyzing large-scale or long-term visibility data, and when analyzing local visibility data with large changes in visibility within a short period of time.

2. Compared to the arithmetic average visibility, the harmonic average visibility can better represent changes in average atmospheric extinction coefficients and average aerosol concentrations. Therefore, we recommend preferentially using harmonic average visibility when calculating the average of visibility data in research related to climate change, atmospheric radiation, atmospheric pollution, environmental health, etc.".

**General Comments:** Overall, I am still not convinced by the authors' claim that the previous methods were wrong and that their method is the only correct method. I still believe that this paper only provides an additional method to calculate visibility, nothing more.

**General Response:** Many thanks again for the comments and suggestions, which help us to restrict our focus to the scientific issues, and avoid pointless arguments about the correctness of the previously used and the proposed method.

Response to reviewers for manuscript AMT-2022-74: **Average visibility that has been miscalculated**

We appreciate the editorial team and the reviewers for their time and comments towards improving our manuscript. We have revised the manuscript carefully, and we respond to all of the reviewers' points below. Responses are given in red.

**Anonymous Referee #2**

**General Comments:**

Zhang et al. report a reassessment of the equations used to calculate the average visibility and propose that the usual understanding of average visibility be dropped in favour of a weighted average. Their work highlights the important connection between the visibility and the underlying atmospheric extinction coefficient, which itself is connected to the composition of the atmosphere (particle concentration, composition, hygroscopicity, etc). Users of visibility data need to be alert to the differences between Eqs. (2) and (3), and that average visibility cannot be straightforwardly related to average extinction coefficient.

The authors do well to bring this to the attention of the community, but their approach has several flaws that should be addressed. I do not think that the work is publishable in its present form. It is not simply a question of the title, an issue raised by the two reviewers. The authors need to revise and clarify the argument and points made in the manuscript.

**General response:** Many thanks to the reviewer for the helpful comments. This time we have made substantial changes to the manuscript to address the reviewers' comments. Specifically, we restrict our focus to proposing a new method for calculating average visibility, rather than dwelling on proving that the old method is incorrect and the new method is correct. We have revised the title, the structure, the argumentative process, and the conclusion of the manuscript. Due to the number of changes, we did not highlight the revised sentences with a different colour. Detailed responses to the comments are below.

**1. Comment 1:** A major objection is that "average visibility" (Eq. 2) conforms to the accepted definition of arithmetic average or arithmetic mean of a given property, and this is what most scientists would understand by the term. The authors are right that Eq. (2) and (3) are not mathematically equivalent. What is needed then is for the authors to provide helpful terminology distinguishing between Eq. (2) ("average visibility") and Eq. (3). There are presumably circumstances when researchers would find the average visibility (Eq. 2) a useful concept for describing their observations. In other cases, a weighted average or some other statistic would be more appropriate (using Eq. (3) to study the underlying extinction coefficient). The circumstances for using one or other statistic needs to be clarified.

Because it may be reasonable to use one or other equation depending on a study's goals, the manuscript's terminology denoting logical conclusions ("therefore", "proves"), and correctness ("correct", "miscalculated", "error") is too narrow. This paper would be much more valuable to the scientific community if it brought greater clarity and nuance

to the ways in which visibility data are analysed.

**Response :** The key question here is whether calculating the average of visibility data is a mathematical or physical problem. We have always considered it a physical problem. And we think a physical problem should have its physical meaning and corresponding process.

The previously used method carries out the summation over the visibility when calculating the average of visibility data. The visibility is neither an extensive quantity nor an intensity quantity, so that the results of the summation of visibility are just numerical values with no corresponding physical process. Therefore, the arithmetic average visibility calculated by the previously used method has no real physical meaning, leading to problems in the average extinction coefficient obtained using arithmetic average visibility. The revised manuscript elaborates on this point. On this basis, the new calculation method is justified from the point of view of physical processes, and the argument continues to explain why the harmonic average visibility calculated by the new method better reflects the changes in average atmospheric extinction coefficients and aerosol concentrations.

We agree that if the purpose is to numerically describe the measured visibility data, then the calculation of average visibility can be treated as a mathematical problem; therefore, it cannot be simply assumed that the previously used method is "incorrect". So the expressions "correct" or "incorrect" are not used in the revised manuscript. We turn to emphasize that the new method is more appropriate for calculating the average visibility when the average visibility is used as a parameter to characterize changes in atmospheric extinction coefficients and aerosol concentrations.

**2. Comment 2:** More emphasis should be placed on when the values produced by the two equations differ. The values produced by Eq. (2) and (3) converge as the range of visibility values becomes increasingly narrow. As Fig. 1c shows, the variation in the results of the two approaches within the hourly dataset is small, presumably because visibility is generally changing little over this time period for most observations. Greater variation occurs when datasets are long enough to contain larger variations in the individual visibility measurements.

**Response:** Thanks for pointing out this. The section describing the relative deviation has been moved to an earlier position to illustrate the differences between the two methods. We strengthen our analysis on the course of the difference, and emphasize that it is important to select the appropriate method for calculating the average visibility when analyzing large-scale or long-term visibility data, and when analyzing local visibility data with large changes in visibility within a short period of time.

**3. Comment 3:** The manuscript's arguments about the average value becoming infinite if any one of the measurement series is infinite are not convincing. Such contributions to the average visibility (or another physical property) do not occur in practice because individual measurements giving infinity are physically implausible (whether for visibility or another physical property) and would be removed from the data set. The formal mathematical possibility of an infinite result is not helpful.

**Response :** We agree that data at infinity can be used as a clue to a problem, but not as a basis for proof. We have removed the analysis of data at infinity and strengthened the theoretical analysis of the physical processes, giving a more general view of the conclusions.

**4. Comment 4:** The manuscript title is uninformative and potentially misleading. Both reviewers thoroughly disliked the title, but the authors were resistant to changing it. I think they should. Perhaps something like "Average visibility and its relationship to atmospheric extinction: a clarification" provides a better summary of their paper's content and aims.

**Response:** Thanks for this comment. Suggested changes have been implemented in the manuscript, where the title has been changed to "
[revised manuscript text omitted]

---

## Author Response (AR3)

Response to reviewers for manuscript AMT-2022-74: **Average visibility that has been miscalculated**

We thank the editorial team and the reviewers for your valuable input in enhancing the quality of our manuscript. We are happy to submit our point-by-point responses to the reviewers' comments and suggestions. The reviewers' comments/suggestions are in black. Our responses are in red. All the suggested changes have been incorporated in the manuscript.

**Anonymous Referee #1**

**General Comments:**
Zhang et al. address the question of the calculation and interpretation of the average visibility and its relationship to the extinction coefficient. While the equation relating visibility and extinction is straightforward, the use of a simple arithmetic average of visibility can be misleading, producing a statistic that is disconnected from the underlying physical cause of visibility reduction, namely, atmospheric extinction. Researchers therefore need to be aware of these considerations if they use visibility datasets to investigate underlying trends in the atmospheric extinction.

This is a revised manuscript that addresses many of the deficiencies of the earlier submission. There some weaknesses in the manuscript that should be addressed to improve its clarity. Some of these are minor textual changes while others require greater clarification. These are addressed below.

**General Response:** We thank the reviewer for the valuable review and encouraging comments.

L1.: Omit colon in title

Thank you for this valuable comment. We agree with the reviewer and have changed our title.

L18: "On the ONE hand"

Thank you for pointing out this error. The suggested change has been implemented in the manuscript. We have also corrected other grammatical errors in the paper.

L32: Rephrase "which greatly supports many research. " Unclear.

Thanks for pointing out this. We now state the following in Line 32"A large amount

of gridded visibility data have been accumulated through long-term observations at dense measurement sites (Pitchford et al., 2007; Singh et al., 2017). These visibility data are widely used and greatly support many research. "

L34: Preferable: "Methodological issues in calculating the average visibility…" or similar

The suggested change has been implemented in the manuscript.

L43: Given that the paper focuses on different average statistics, these should be specified explicitly…. "The second method first calculates the arithmetic average of the extinction coefficient, ". And likewise elsewhere in the manuscript.

Suggested changes have been implemented in the manuscript; where "the average extinction coefficient" has changed to "the arithmetic average of the extinction coefficient" in this paragraph. (Line 44- Line 51)

L61: The sentence "The answer seems clear, but not yet convincing. " adds little to the argument and should be dropped.

This sentence has been removed from the manuscript.

L65: Better: "been overlooked in the past. "

The suggested change has been implemented in the manuscript.

L68: "If the difference…is also reliable". The authors have already pointed out that arithmetically the two methods are not the same, while "reliability" relates to the purpose that the data is being used for and the accuracy and precision needed. Some clarification is needed here.

We thank the reviewer for this useful comment. The sentence has been modified as follows in Line 70: "If the difference is negligible, there is no point in discussing this issue, and the arithmetic average visibility obtained from Eq. 2 can be used with small error. "

L74: It seems what the authors are doing is not developing an "intuitive understanding", but rather demonstration the divergence of the two approaches with a

given dataset.

We have revised the sentence, which now reads "To visualize the magnitude of the numerical difference between arithmetic average visibility and harmonic average visibility, …"

Fig. 1: It is difficult to compare the short term effects and the two different methods for calculating the average visibility since they are on separate graphs. It would be helpful to have plots (e.g., monthly, yearly) on the same figure to improve comparison.

We appreciate the reviewer's suggestion. We think that Fig. 1 has achieved our intentions, and no changes have been made. The main focus of Fig. 1a and Fig. 1b is to show the average visibility data calculated by the two methods. The comparison of them is shown in Fig. 1c, which illustrates the distribution of the relative deviation of arithmetic average visibility from harmonic average visibility. Figure 1c shows that the relative deviation of the yearly average is larger than that of the monthly average, which is larger than that of the hourly average. It can be drawn from Fig. 1c that the difference between the two calculation methods cannot be ignored, especially when analyzing large-scale or long-term visibility data, and when analyzing local visibility data with large changes within a short period of time. No changes have been made.
The relative deviation is not negligible, so that the difference between the two calculation methods cannot be ignored, especially when analyzing large-scale or long-term visibility data, and when analyzing local visibility data with large changes within a short period of time. No changes have been made.

L104: Correct "period CHOSEN to"

The suggested change has been implemented in the manuscript.

L143: I presume "extensive" property is what is meant?

Yes. Extensive property is a property of a component or system that is a function of the whole component, a property that changes if material is added or subtracted to the component ( https://en.wiktionary.org/wiki/extensive_property). Extensive properties are proportional to the amount of matter in the system, while intensive properties do not depend on the size of the system, nor the amount present in the system.

L146: "the summation of visibility has no real physical meaning". That might be so, yet it could still be a useful statistic.

Thanks for pointing out this. We now state the following in Line 146: "Therefore, the summation of visibility data is just a useful statistic without real physical meaning. "

L147: "3.2 Physical meaning of arithmetic average visibility and harmonic average visibility" is a clearer and shorter section title.

The suggested change has been implemented in the manuscript.

L154/5: This section is confusing as $M\_j$ here relates to the average mass extinction of the SAMPLE, not individual SUBSTANCES composing the sample, which is the meaning for the same symbol in Table 1. This should be clarified in the text and the properties and symbols distinguished from one another.

We appreciate the constructive comment made by the reviewer. The word "substances" is changed to "components" in section 3.1, and the word "substances" is changed to "samples" in section 3.2.

L158: Change "substances" to "sample" as in the above.

The suggested change has been implemented in the manuscript.

L183: Perhaps "property" is better than "process".

The word "process" is changed to the word "meaning". The sentence now reads "…so that the results of the summation of visibility are just numerical values with no corresponding physical meaning" (Line 183)

L197: I think this is a statistical rather than a "mathematical" problem.

The suggested change has been implemented in the manuscript.